# Numerical Simulation of Mixing Fluid with Ferrofluid in a Magnetic Field Using the Meshless SPH Method

Mohsen Abdolahzadeh [1], Ali Tayebi [2], Mehrdad Ahmadinejad [2] and Božidar Šarler [1,3,*]

1 Faculty of Mechanical Engineering, University of Ljubljana, Aškerčeva cesta 6, 1000 Ljubljana, Slovenia
2 Department of Mechanical Engineering, Yasouj University, Yasouj 75918-74831, Iran
3 Institute of Metals and Technology, Lepi pot 11, 1000 Ljubljana, Slovenia
* Correspondence: bozidar.sarler@fs.uni-lj.si

**Abstract:** In this study, a numerical investigation of the effect of different magnetic fields on ferrofluid-fluid mixing processes in a two-dimensional microchannel is performed An improved version of smoothed particle hydrodynamics, SPH, by shifting particle algorithm and dummy particle boundary condition, is implemented to solve numerical continuity, ferrohydrodynamics-based momentum and mass transfer equations. SPH is formulated through the irregular arrangement of the nodes where the fields are approximated using the fifth-order Wendland kernel function. After validating the computational approach, the influence of the number (from one to three) of parallel electrical wires positioned perpendicular to the microchannel on the mixing efficiency is studied for the first time. It has originally been found that the mixing efficiency highly non-linearly depends on the Reynolds number and the number of electrical wires. For $Re \leq 20$, the mixing efficiency is almost the same for two and three electrical wires and about two times higher than one electrical wire. For $Re \geq 80$, the mixing efficiency of three wires is much higher than one and two electrical wires. Optimum performance of the micromixer is achieved with three electrical wires, since the mixer performs well on a broader range of $Re$ than the other two studied cases. The outcomes of this study, obtained by a meshless method, are important for the industrial design of micromixers.

**Keywords:** mixing process; ferrofluid; magnetic field; active micromixer; SPH

## 1. Introduction

The study of the mixing process of fluids in the submillimeter scales represents an interesting research subject due to the design of different microfluidic systems. Micromixing is the process of mixing at the smallest scale of fluid motion (i.e., molecular scale). Micromixing has various applications such as in microfluidic lab-on-a-chip devices, drug delivery, bioengineering, etc. [1]. There are two general types of micromixers: passive and active [2]. Passive micromixers do not need any external force for flow disturbance. In this case, the mixing process is governed primarily by diffusion and convection due to the flow's kinetic energy. Active micromixers, on the other hand, require an external force for disturbance. Although active micromixers, compared to passive ones, are often more complex and expensive, they are essential in some of applications [3]. There are different external actuators to improve the mixing process, including acoustic field [4–6], thermal field [7,8], pressure-driven field [9–15], electric field [16–19], and magnetic field [20]. In the present work, the effect of the magnetic field on the mixing process in a channel flow will be explored. In the following, literature on the micromixing process, focused on the magnetic field, is reviewed.

Unlike the macroscale fluidic devices, where the mixing process mainly depends on the convection mechanism, mixing at the microscale mostly depends on diffusion. Moreover, due to the microscale system's small dimensions and the fluid's low velocity, the Reynolds number is small, and thus, the flow regime is laminar [21].

Andersson et al. [22] have numerically explored the effect of the magnetic field on a viscous ferrofluid and concluded that the presence of the magnetic field increases the

skin friction. The biomagnetic fluid flow over a stretching sheet, considering a non-linear temperature-dependent magnetization, is studied in Ref. [23]. It is often assumed that the magnetic field strength is adequately strong to make the biofluid saturated, and also, the magnetization can be stated as a function of both the magnetic field intensity and temperature [24,25]. Tzirtzilakis [26] proposed a mathematical model for biomagnetic fluid dynamics to describe a laminar incompressible Newtonian blood flow under the influence of a magnetic field. He also studied the effects of the strength and the gradient of the magnetic field on the flow field properties. Tsai et al. [27] experimentally showed that by a well-positioned magnet, the mixing performance of a ferro-nanofluid and water can be improved. The mixing mechanisms of a rapid microfluidic mixer actuated by direct or alternating currents of electricity (DC or AC) have been investigated in Ref. [28]. A numerical simulation of an active mixing system has been examined in Ref. [29] by applying a hybrid gradient of a magnetic field composed of a combination of a static gradient magnetic field and an external uniform magnetic field of alternating current. The effects of the magnetic field on fluid flow and heat transfer in an aneurysm were investigated numerically by Sharifi et al. [30].

The mixing of Newtonian and non-Newtonian fluids in a three-dimensional steady-state micromixer, actuated with a magnetic field [31] and in a spiral passive micromixer [32], has been explored numerically. Lab-on-chip devices, one of the micromixers applications, were investigated by studying the effects of steady and periodic magnetic fields on their mixing performance [33]. Furthermore, magnetic drug targeting, considering the impact of a magnetic field, was introduced (e.g., see Refs. [34,35]). Many other micromixers, actuated by a magnetic field, have been presented in the literature. See reviews on micromixers with magnetic actuators [36–38] for more details.

The meshless methods, particularly the particle-based category, have recently received increased attention from researchers studying the mixing process [39–42]. Various types of meshless approaches, such as Moving Least Square (MLS) [43,44], particle methods [45], local radial basis function collocation method [46,47], and many others, can be found in Ref. [48]. Smoothed particle hydrodynamics (SPH), a fully Lagrangian particle-based method, was first conducted to model astronomical phenomena [49,50]. Since then, this method has been developed and applied for different fluid dynamics situations. In particular, to properly simulate problems such as free surface flows [51–53], multiphase flows [54–56], and other pertinent subjects. Due to the various applications of the mixing process and its importance, the SPH method is utilized here to study the micromixing performance under the influence of the magnetic field. Therefore, the main goal of the current work is to numerically investigate the mixing process of a laminar Newtonian viscous fluid and a ferrofluid in the presence of a differently configured magnetic field. By considering a simple relation between the magnetization M and the magnetic field strength H, the influence of the number of different conducting wires, positioned perpendicular to the microchannel on the mixing index is investigated.

The remainder of the paper is organized as follows; Section 2 describes the geometry of the micromixers. Section 3 presents the governing equations. Section 4 contains the numerical procedure. In Section 5, the results are discussed; finally, in Section 6, the conclusions are presented.

## 2. Physical Models Description

A scheme of the considered micromixer is depicted in Figure 1. The microchannel depth is assumed to be large enough to ignore variations of the flow field properties perpendicular to the x-y plane. Several experimental studies proved the possibility of studying the mixing process in two dimensions [57,58]. Therefore, considering 2D models of micromixers provides a reasonably good approximation to some of their designs [59]. Dimensions of microchannel here are specified based on a unit length, *UL*. As Figure 1 shows, the two fluids enter the microchannel with the same velocity, $U_{in}$. The miscible fluids have the same viscosity and density, whereas their concentrations are $C_o = 0.0$ and

$C_1 = 1.0$, at the entrance of the microchannel. The three electrical wires generating the magnetic field are located at fixed positions, as depicted in (Figure 1).

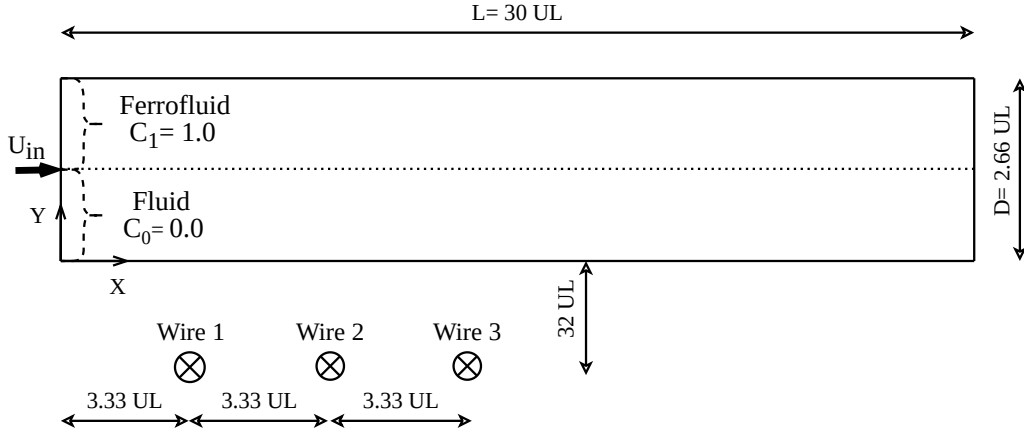

**Figure 1.** Schematic of the physical models of the micromixers.

### 3. Governing Equations

In this section, the conservation laws, i.e., the continuity, momentum, and mass transport equations, as the governing equations to study the effects of different magnetic fields on the mixing process of ferrofluid, using a mixing index defined subsequently, are presented as:

$$\frac{1}{\rho}\frac{\partial \rho}{\partial t} + \frac{\partial u_i}{\partial x_i} = 0, \tag{1}$$

$$\frac{\partial u_i}{\partial t} + u_i\frac{\partial u_j}{\partial x_i} = -\frac{1}{\rho}\frac{\partial p}{\partial x_i} + \nu\frac{\partial^2 u_j}{\partial x_i \partial x_i} + M_n H\frac{\partial H}{\partial x_i}, \tag{2}$$

$$\frac{\partial C}{\partial t} + u_i\frac{\partial C}{\partial x_i} = \alpha_c\frac{\partial^2 C}{\partial x_i \partial x_i}, \tag{3}$$

where $t$, $x_i$, $\rho$, $u_i$, $p$, $\nu$, $M_n$, $H$, $C$, and $\alpha_c$ are the time, direction of the 2D Cartesian system (here are X and Y), the density of the fluid, velocity components, pressure, kinematic viscosity, dimensionless magnetic number, the magnitude of the magnetic field, concentration, and mass diffusivity, respectively. The last term on the right-hand side of Equation (2) is the magnetic force. This form of the magnetic term can be used when the assumption of $\vec{M} \ll \vec{H}$ is met [26]. The relation between the magnetization $M$ and the magnetic field intensity $H$ is stated as $\vec{M} = \chi\vec{H}$, where $\vec{M}$ and $\chi$ are the magnetization and the magnetic susceptibility, respectively. Considering the Biot–Savart law, the components of the magnetic field $\vec{H}$ in the directions of $x$ and $y$ can be written as [25]:

$$H_x(x,y) = \frac{I}{2\pi}\frac{1}{(x-a)^2 + (y-b)^2}(y-b), \tag{4}$$

$$H_y(x,y) = \frac{I}{2\pi}\frac{1}{(x-a)^2 + (y-b)^2}(x-a), \tag{5}$$

where $a$ and $b$ are the $x$ and $y$ positions of the wires, respectively. Thus, the magnetic field intensity as a function of the electric current $I$ can be determined as:

$$H(x,y) = \sqrt{H_x^2 + H_y^2} = \frac{I}{2\pi}\frac{1}{\sqrt{(x-a)^2 + (y-b)^2}}. \tag{6}$$

The dimensionless magnetic number $M_n$ is defined as:

$$M_n = \frac{D^2\mu_0\chi H_o^2}{\nu^2\rho} = \frac{D^2 B_0 M_0}{\nu^2\rho}, \tag{7}$$

where $\mu_0 = 4\pi \times 10^{-7} N/A^2$ is the magnetic permeability of the vacuum and $D$ is the width of the channel shown in Figure 1. In addition, $H_0$, $B_0 = \mu_0 H_o$ and $M_0 = \chi H_o$ are respectively the magnetic strength, the magnetic field induction, and the magnetization in the minimum distance between the lower wall of the microchannel and the location of the wires.

To close the listed governing equations, an equation of state is required:

$$P = c_0{}^2(\rho - \rho_0), \tag{8}$$

in which $c_0$ and $\rho_0$ are the speed of sound within the fluid and initial density, respectively. The speed of sound is assumed to be much higher than the maximum fluid velocity. This assumption is made to decrease the fluctuations of the density. The no-slip boundary condition for velocity and the Neumann boundary condition for pressure and concentration at walls are considered.

To quantify the mixing performance of micromixers, a mixing index, based on the standard deviation of concentration, can be defined as [3]:

$$M_I = \sqrt{\frac{1}{N} \sum_{j=1}^{N} (\frac{C_j - C_{mean}}{C_{mean}})^2}, \tag{9}$$

where $N$, $C_j$, and $C_{mean}$ are, respectively, the total number of particles, the concentration of particle $j$, and the mean concentration (here $C_{mean} = 0.5$). The mixing index can be evaluated from 0, unmixed state, to 1, full mixed state. Equation (15) can be used to represent the mixing efficiency as [3]:

$$\eta_{mix} = (1 - M_I) \times 100. \tag{10}$$

Higher values of the mixing efficiency $\eta_{mix}$ indicate greater homogeneity of the mixed fluids and thus higher performance of the micromixer.

## 4. Numerical Procedure

As mentioned before, SPH is one of the frequently used Lagrangian meshless methods for modeling complex phenomena in physics. The method is based on particle theory, in which continuous partial differential equations are approximated by algebraic equations. A comprehensive description of the theory and applications of SPH can be found in Ref. [60].

The function of interpolation for a scalar field $A$, the pressure and concentration, or a vector field $\mathbf{A}$, the velocity and magnetic field, of particle $a$, which is located at position $r_a$, in the SPH framework are defined respectively as [61]:

$$\{A_a\} \approx \sum_b V_b A_b w_{ab}, \tag{11}$$

$$\{\mathbf{A_a}\} \approx \sum_b V_b \mathbf{A_b} w_{ab}, \tag{12}$$

in which $V_b$ is the volume of particle $b$, $w_{ab} = w_h(\mathbf{r_a} - \mathbf{r_b})$ is the function of interpolation and $h$ is the smoothing length. The Wendland kernel function of 5th order is considered in the present work [62]:

$$w_h(q) = \frac{\alpha_w}{h^2} \begin{cases} (1 - \frac{q}{2})^4(2q + 1) & 0 \leq q \leq 2 \\ 0 & 2 \leq q, \end{cases} \tag{13}$$

where $\alpha_w = {}^7/_{4\pi}$ is a constant, $q = \frac{|\mathbf{r_a} - \mathbf{r_b}|}{h}$, $\mathbf{r_a}$ and $\mathbf{r_b}$ are the position vectors of particles $a$ and $b$, respectively.

The operators such as gradient (**grad**), divergence (div), and laplacian (div(**grad**)) for a scalar (*A*) and a tensor (**A**) are expressed as follows [61,63]:

$$\{(\mathbf{grad}\,A)_a\} \approx \sum_b V_b A_{ba} \mathbf{B_a}.\acute{w}_{ab},$$
(14)

$$\{(\mathrm{div}\,\mathbf{A})_a\} \approx \sum_b V_b \mathbf{A}_{ba}(\mathbf{B_a}.\acute{w}_{ab}),$$
(15)

$$\mathrm{div}\{\mathbf{grad}\{A_a\}\} \approx \sum_b 2V_b \frac{A_{ba}}{r_{ab}} \mathbf{e_{ab}}.(\mathbf{B_a}.\acute{w}_{ab}),$$
(16)

in which $\mathbf{e_{ab}}$, $\acute{w}_{ab}$, and $\mathbf{B_a}$ are respectively the unit vector between particles a and b, the derivative of the kernel function, and the factor of renormalization, which is determined as [61]:

$$\mathbf{B_a} = -[\sum_b V_b(\mathbf{r}_{ab})\acute{w}_{ab}]^{-1}.$$
(17)

Implementation of boundary conditions represents a challenge in the SPH method. Various techniques have been developed to treat the boundary conditions properly [62–64]. In this work, the Dummy particle method [65] is used for wall boundaries, and the open channel flow conditions are considered for inflow and outflow boundaries [64]. For further information on these techniques, see the work of Lee et al. [65]. Different remedies were also proposed for solving the non-uniformity distribution in SPH [66,67]. We have used the particle shifting algorithm for this purpose [68]. Furthermore, the prediction-correction scheme is used for the time stepping algorithm, in which a variable time-step size is utilized as [69]:

$$\Delta t = \varepsilon_t \min(\frac{\delta_{\min}}{c + V_{\max}}, \frac{\delta^2_{\min}}{\nu}, \sqrt{\frac{\delta_{\min}}{g}}, \frac{\delta^2_{\min}}{\alpha_c}),$$
(18)

where $\epsilon_t$ is a constant ($0 < \varepsilon_t < 1$), $\delta_{\min}$ is the minimum distance between two particles, $g$ is gravity acceleration and $V_{\max}$ is the maximum particle velocity.

## 5. Results and Discussions

### 5.1. Validation

In order to verify and check the accuracy of the computational approach, the problem of the mixing process of two fluids in a rectangular microchannel is solved and the results are compared to those of Kim et al. [70], which were obtained by the Lattice Boltzmann method (LBM). LBM, which originated in lattice gas automata, is another class of computational fluid mechanics that uses streaming and collision (relaxation) processes to simulate a fluid density on a lattice rather than directly implementing the Navier–Stokes equations. In contrast, SPH is a meshless method that directly solves the Navier–Stokes equations to simulate fluid flow in a Lagrangian framework. The Reynolds number $Re = 240$ and the Schmidt number $Sc = 10$, defined as the ratio of the kinematic viscosity to the mass diffusivity, are the characteristic dimensionless numbers of this problem. A total of 14,000 computational particles are used here to simulate the problem. Figure 2 shows the variations of the concentration versus the non-dimensional characteristic length of the channel. A good agreement between the LBM and SPH results is observed.

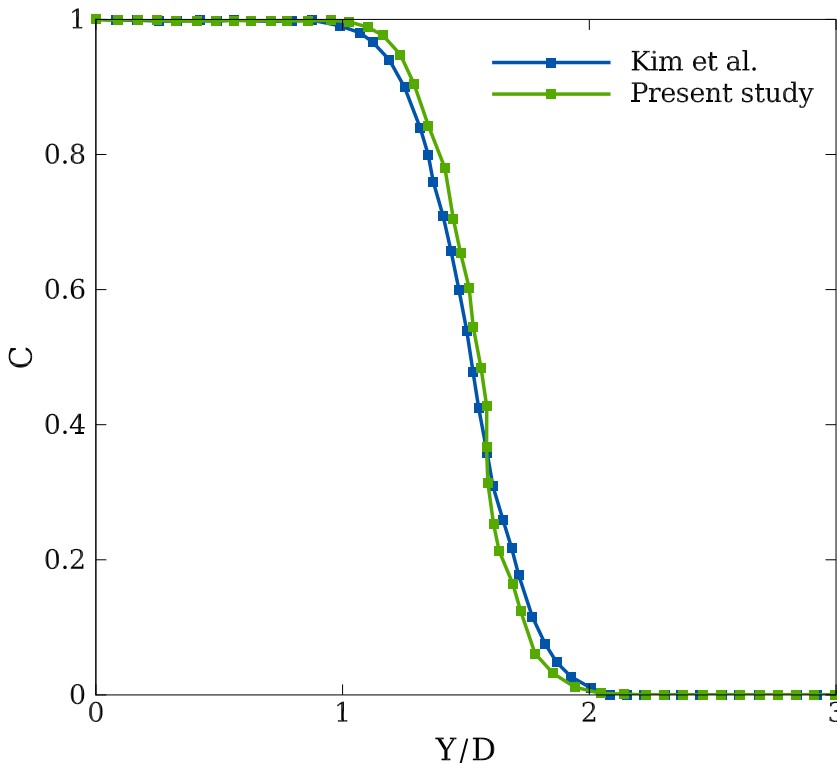

**Figure 2.** Comparison of the variation of concentration versus the dimensionless length of the channel in the work of Kim et al. [70].

*5.2. Mixing Process under the Effect of Magnetic Fields*

The results of the numerical simulation, using the SPH method, for the mixing process of ferrofluid under the influence of different magnetic fields follow. This arrangement contributes to the forced mixing due to the external magnetic field force. For the problem under consideration, the magnetic number $M_n = 1000$, saturation magnetization $M_s = 60\, A/m$, magnetization at the location of wires $M_o = 60$, Schmidt number $Sc = 1$ and different Reynolds numbers, $1 \leq Re \leq 100$, are considered. Furthermore, the effects of different numbers of electrical wires with various Reynolds numbers are explored; to this end, the contours of velocity, concentration, and particle distribution for three electrical wires at $Re = 15$ are drawn in Figures 3 and 4. It is worth noting that the results of $Re = 15$ are presented here as a sample of the results of the other Reynolds numbers. Figure 3 suggests that by increasing the number of wires, the production of the vortices also increases. The vortex shedding phenomenon leads to an increase in the interface between two fluids and ferrofluid, and subsequently, the mixing efficiency is enhanced. The main reason for this enhancement is due to the increased contact time of the fluids due to the vortex shedding. Figure 4 indicates that an increase in the number of wires leads to a better homogeneity for the fluid in a specific desired time. It is worth mentioning that the results are presented at $t^* = 0.014$, where $t^* = tU_{in}/UL$ is the dimensionless time. Moreover, considering both the contours of concentration and particle distribution, the particle dispersion at the outlet of the channel is improved, which favorably influences the mixing index. The mixing efficiencies at $Re = 15$ for one, two, and three wires are 61.85%, 72.22%, and 77.16%, respectively, confirming this fact.

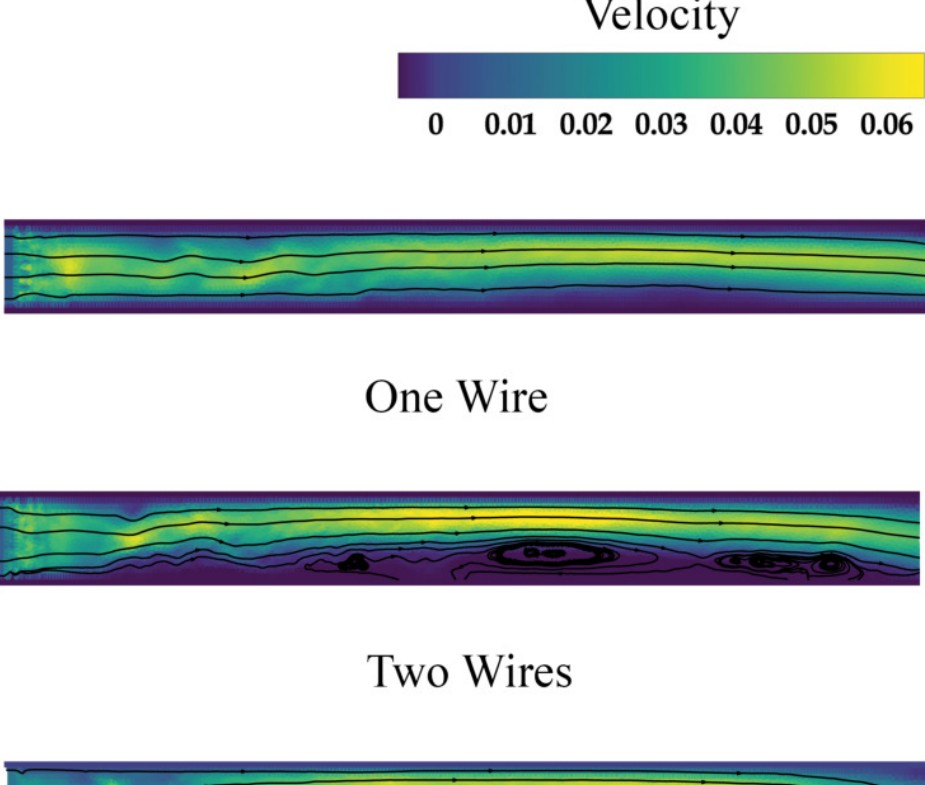

**Figure 3.** Velocity contours for different number of wires at $Re = 15$.

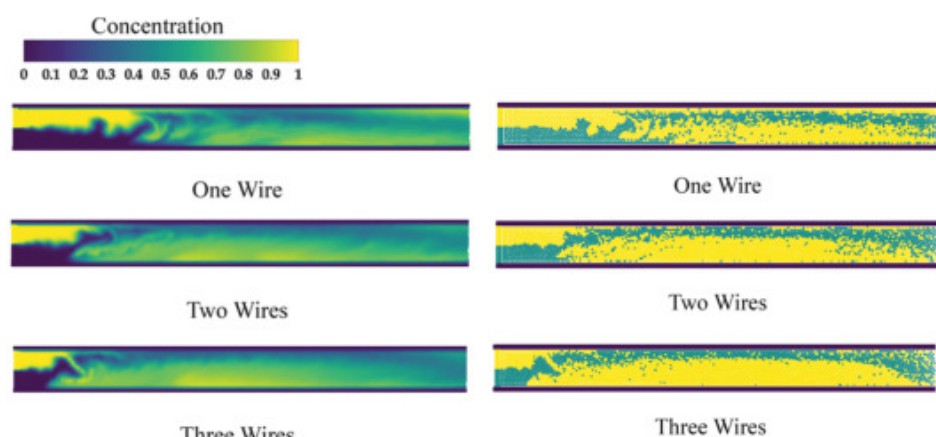

**Figure 4.** Concentration (**left**) and particle contours (**right**) for a different number of wires at $Re = 15$.

Figure 5 shows a comprehensive view of the effects of the number of electrical wires on the mixing efficiency in a wide range of Reynolds numbers. This figure can be approximately divided into some small intervals; for the interval of $0 \leq Re \leq 20$, the mixing efficiencies for the case of two and three wires are better than in the case of one wire; moreover, the three-wires case has higher efficiencies, compared to the case of two wires. This trend can be interpreted as the vortex shedding intensification due to an increase in the number of wires so that the two fluid elements are more in contact and have more time to become homogenized. Interestingly, there is an intersection point around $Re = 20$ for all

three cases. In the interval of $20 \leq Re \leq 52$, however, the effect of the number of electrical wires is opposed to that of the beginning interval; furthermore, the behavior of the one wire case is almost parabolic, and surprisingly this case has better efficiencies, compared to the other cases. In addition, it is indicated that the performance of the three-wires case is the worst, in this interval. As the Reynolds number exceeds 55, the mixing efficiency of the one wire case significantly decreases. In contrast, this behaviour for the case of two wires is observed at the Reynolds number of about 70. The three-wires case performance slightly increases in the interval of $60 \leq Re \leq 100$. For Reynolds numbers higher than 80, the mixing performance of the three-wires case increases compared to the two other cases. The overall result indicates that the number of electrical wires positively affects the performance of the micromixing process. However, it is fair to say that the mixing efficiency for the case of three wires has a more stable behavior (fewer ups and downs) for different Reynolds numbers. If one tends to consider a micromixing device working in a wide range of Reynolds numbers, it is more suitable to use the three electrical wire case.

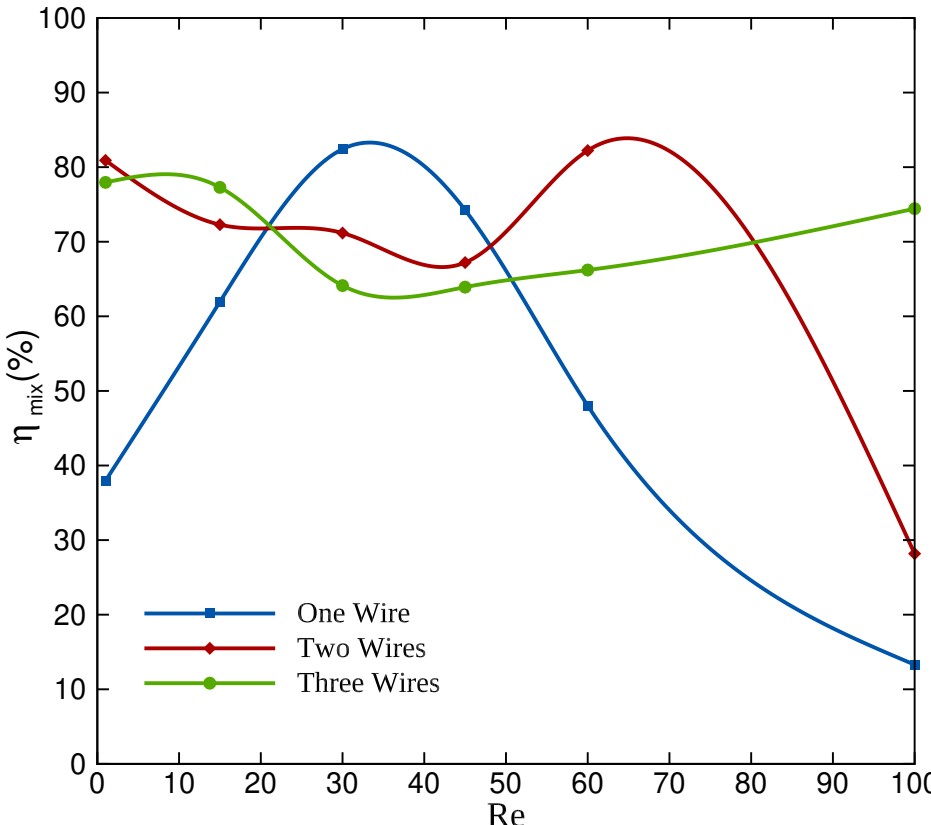

**Figure 5.** Comparison of the mixing efficiency versus the Reynolds numbers for a different number of electrical wires.

Figure 6 shows the variation of the mixing index with the dimensionless length of the microchannel for different Reynolds numbers; from which it can be concluded that for the cases of $Re = 15, 30, 45$, the effect of the number of electrical wires at the end of the microchannel is less significant, compared to the other Reynolds numbers. For the case of $Re = 1$, the influence of increasing the number of electrical wires on the mixing performance is more effective, in comparison with the other Reynolds numbers, at the end of the microchannel (Figure 6). The mixing index values at $Re = 100$ for the cases of one and two wires are almost identical, except at the end of the channel, whereas the three-wires case clearly indicates a significant improvement in the mixing process (Figure 6).

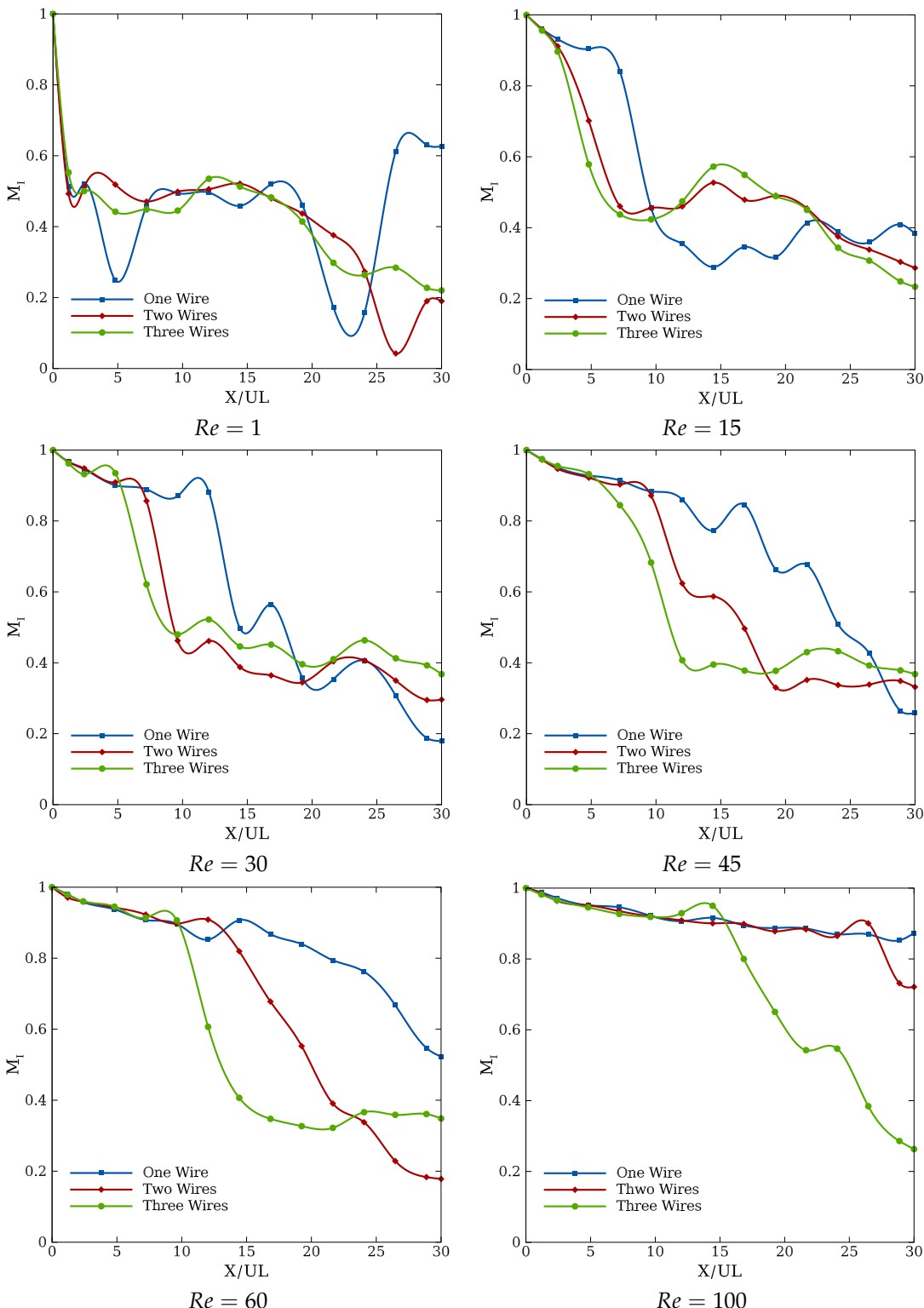

**Figure 6.** Variations of the mixing index versus the dimensionless length of the microchannel for various Reynolds numbers.

Generally speaking, for a wide range of Reynolds numbers, the case of a micromixer with three wires is optimal, among all the examined cases. The three-wires case maintains the behavior of mixing efficiency at a high level for different Reynolds numbers.

## 6. Conclusions

In the present work, the mixing process of ferrofluid in a microchannel was studied numerically using the meshless SPH method. The effects of increasing the number of magnetic fields on the mixing index for a wide range of applicable Reynolds numbers have been explored. After validating the computational approach, the results for the problem under consideration were presented. The most important findings of the current investigation can be summarized as:

- By increasing the number of electrical wires, for some Reynolds numbers, the vortex shedding intensifies and causes an increase in the interface between the two fluids, which leads to a better mixing efficiency;
- The homogeneity of the mixing is improved by increasing the number of wires;
- The micromixer with three wires has optimum performance for a range of Reynolds numbers compared to the cases of one and two wires;
- For $Re = 15, 30, 45$, the variation of efficiency for different numbers of wires is not as significant as that of the other Reynolds numbers at the outlet of the channel;
- If one aims to design a micromixer working in a wide range of Reynolds numbers, the results show that the case with three magnetic wires is recommended.

The findings of the represented study, performed by a meshless method, are important for the industrial design of micromixers. The future extension of this study will focus on mixing non-Newtonian fluids in domains with geometrically moving boundaries and channels with passive mixers, for which SPH is the most suitable approach. This research also represents the power of meshless methods in industrial applications.

**Author Contributions:** Conceptualization, M.A. (Mohsen Abdolahzadeh) and A.T.; validation, M.A.(Mohsen Abdolahzadeh); formal analysis, M.A. (Mohsen Abdolahzadeh) and A.T.; resources, M.A.(Mohsen Abdolahzadeh); data curation, M.A.(Mohsen Abdolahzadeh); writing—original draft preparation, M.A.(Mohsen Abdolahzadeh), A.T., and M.A.(Mehrdad Ahmadinejad); writing—review and editing, A.T. and B.Š.; supervision, B.Š.; project administration, B.Š.; funding acquisition, B.Š. All authors have read and agreed to the published version of the manuscript.

**Funding:** This research was made possible by support from Yasouj University and University of Ljubljana under core funding P2-0162 and project J2-4477.

**Institutional Review Board Statement:** Not applicable.

**Informed Consent Statement:** Not applicable.

**Data Availability Statement:** The data presented in this study are available on request from the corresponding author.

**Acknowledgments:** The first author is grateful for the support from the Slovenian Grant Agency ARRS in the framework of the Young Researcher Program.

**Conflicts of Interest:** The authors declare no conflict of interest.

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
