# Peer review of "Numerical Simulation of Mixing Fluid with Ferrofluid in a Magnetic Field Using the Meshless SPH Method"

_fluids, doi:10.3390/fluids7110341_

Round 1

Reviewer 1 Report

The impact of three different magnetic fields due to three wires on ferrofluid-fluid mixing processes in a two-dimensional microchannel is analyzed numerically. The most important result of this paper is the mixing efficiency analysis for increasing the number of electrical wires for different Reynolds numbers for the two types of fluids (a ferrofluid and a fluid).

The authors must look carefully at the below points:

Create development of paper in the introduction part.

Clearly mention the novelty of the paper in the abstract.

The authors should explain the limitations of this work in the introduction section.

Specify the ferrofluid and the fluid used in the analysis (according to Fig.1) Define UL, C_1 and C_0.

What assumptions are applied in mathematical modeling?

Specify C_j and C_mean in equation (9).

Define the coordinate system for the position vectors.

Check alpha in row 138.

Check the citation in row 156.

In case of validation specify the fluid (ferrofluid and fluid) and geometry.

In Fig. 5 eta should be in %.

In Fig. 6 the Reynolds numbers cannot be seen!

What is the accepted mixing efficiency in practice?

How does the mixing efficiency change if the positioning of the wires is changed?

What effect does the magnetic number have on the mixing efficiency?

The authors should explain why the study is useful with a clear statement of novelty or originality by providing relevant information in the introduction and conclusion sections.

References 23, 24 and 27-59 are not cited in the paper.

Some important works related to the numerical methods of ferrofluid flows and their applications should be discussed in the introduction part and be added to the references lists:

G Bognar, K Hriczo, Ferrofluid flow in magnetic field above stretching sheet with suction and injection, Mathematical Modelling and Analysis 25 (3), 461-472, 2020, doi: 10.24352/UB.OVGU-2019-002

S. Nadeem, N. Ullah, A.U. Khan and T. Akbar. Effect of homogeneous- heterogeneous reactions on ferrofluid in the presence of magnetic dipole along a stretching cylinder. Results in Physics, 7:3574-3582, 2017. doi:10.1016/j.rinp.2017.09.006.

J.L. Neuringer. Some viscous flows of a saturated ferrofluid under the combined influence of thermal and magnetic field gradients. International Journal of Non-Linear Mechanics, 1(2):123-137, 1966. doi: 10.1016/0020-7462(66)90025-4.

Author Response

Response to the Reviewers' Comments

Fluids

Manuscript Title: Numerical Simulation of Mixing Fluid with Ferrofluid in a Magnetic Field by Using the Meshless SPH Method

Authors: M. Abdolahzadeh, A. Tayebi, M. Ahmadinejad, and B. Šarler

Manuscript ID: fluids-1947965

Response to reviewer #1:

The authors wish to thank the reviewer for his/her dedication to review our manuscript and for his/her very helpful comments that contributed to the quality of our publication.

All of the reviewers' comments are addressed point by point in the following. Please note that the reviewers' comments are in black, replies to the reviewer are in blue and revisions made to the manuscript are in green.

Remark 1.1: Create development of paper in the introduction part.

Reply 1.1: The first author apologizes for the problem that occurred when running Latex. In the Latex source file, a command, \begin{linenomath*}, caused the converted pdf file not to display part of the manuscript, part of the introduction (section 1) and the physical models description (section 2). the physical description was presented in section 2. Therefore, the latex problem was fixed, and Sections 1 and 2 were added.

Remark 1.2: Clearly mention the novelty of the paper in the abstract.

Reply 1.2: The authors stated their original result in the abstract in line 8. However, the following paragraph was added as another novelty of the numerical method.

Revision 1.2:‌ An improved version of smoothed particle hydrodynamics, SPH, by shifting particle algorithm and dummy particle boundary conditions is implemented to solve numerical continuity, ferro-hydrodynamics based momentum and mass transfer equations.

Remark 1.3: The authors should explain the limitations of this work in the introduction section.

Reply 1.3: The correction has been done in the revised introduction section.

Remark 1.4: Specify the ferrofluid and the fluid used in the analysis (according to Fig.1) Define UL, C_1 and C_0.

Reply 1.4: The correction has been done in the new revision of Section 2.

Remark 1.5: What assumptions are applied in mathematical modeling?

Reply 1.5: A 2D model of micromixers was considered in the simulation, and the reason for the approximation was described in the revision of Section 2 in line 83.

Another assumption is related to the magnetic term when M "H, which was described in Section 3 in line 103.

Remark 1.6: Specify C_j and C_mean in equation (9).

Reply 1.6: The correction has been done.

Remark 1.7: Define the coordinate system for the position vectors.

Reply 1.7: A 2D cartesian coordinate system (X-Y) has been presented in Fig. However; the manuscript has been corrected accordingly.

Revision 1.7:‌ where t, xi, ρ, ui , p, ν, Mn, H, C, and αc are the time, direction of the 2D Cartesian system (here are X and Y), density of the fluid, velocity components, pressure, kinematic viscosity, dimensionless magnetic number, the magnitude of the magnetic field, concentration, and mass diffusivity, respectively.

Remark 1.8: Check alpha in row 138.

Reply 1.8: The authors clarified that the αw is a constant.

Remark 1.9: Check the citation in row 156.

Reply 1.9: Thank you. The related citation has been corrected.

Remark 1.10: In case of validation specify the fluid (ferrofluid and fluid) and geometry.

Reply 1.10: The manuscript has been corrected accordingly.

Revision 1.10:‌ In order to verify and check the accuracy of the computational approach, the problem of the mixing process of two fluids in a rectangular microchannel is solved ...

Remark 1.11: In Fig. 5 eta should be in %.

Reply 1.11: The manuscript has been corrected accordingly.

Remark 1.12: In Fig. 6 the Reynolds numbers cannot be seen!

Reply 1.12: The manuscript has been corrected accordingly.

Remark 1.13: What is the accepted mixing efficiency in practice?

Reply 1.13: The accepted mixing efficiency depends on the Reynolds number and magnetic number. In other words, the authors can not provide a unique mixing efficiency for all Reynolds numbers. For example, at Re=10, the accepted mixing efficiency is about 73%, while at Re = 1 it has another value. Fig. 5 presents the mixing efficiency versus the Reynolds numbers for different numbers of electrical wires.

Remark 1.14: How does the mixing efficiency change if the positioning of the wires is changed?

Reply 1.14: In the present work, the location of the wires was chosen to be the same for all case studies. The number of electric wires and Reynolds number were considered as factors for the study.

Remark 1.15: What effect does the magnetic number have on the mixing efficiency?

Reply 1.15: In the present study, the effect of one, two and three magnet wires on the mixing processes for 0<Re100 was investigated. Fig. 5 shows the effect of the magnet wire on the mixing process for different Reynold numbers.

Remark 1.16: The authors should explain why the study is useful with a clear statement of novelty or originality by providing relevant information in the introduction and conclusion sections.

Reply 1.16: In the first paragraph of the introduction, the authors explained the importance of micromixers in different applications. Therefore, the design of high-performance micromixers is an important issue in this area. The present research sought a high-performance micromixer for mixing ferrofluid-based fluid using different magnetic numbers. In the conclusion section, the authors explain the importance of the current study in the industrial design of micromixers and recommend using the case with three magnetic wires if one aims to design a micromixer working in a wide range of Reynolds numbers.

Remark 1.17: References 23, 24 and 27-59 are not cited in the paper.

Reply 1.17: The manuscript has been corrected accordingly.

Remark 1.18: Some important works related to the numerical methods of ferrofluid flows and their applications should be discussed in the introduction part and be added to the references lists:

G Bognar, K Hriczo, Ferrofluid flow in magnetic field above stretching sheet with suction and injection, Mathematical Modelling and Analysis 25 (3), 461-472, 2020, doi: 10.24352/UB.OVGU-2019-002

S. Nadeem, N. Ullah, A.U. Khan and T. Akbar. Effect of homogeneous- heterogeneous reactions on ferrofluid in the presence of magnetic dipole along a stretching cylinder. Results in Physics, 7:3574-3582, 2017. doi:10.1016/j.rinp.2017.09.006.

J.L. Neuringer. Some viscous flows of a saturated ferrofluid under the combined influence of thermal and magnetic field gradients. International Journal of Non-Linear Mechanics, 1(2):123-137, 1966. doi: 10.1016/0020-7462(66)90025-4.

Reply 1.18: The manuscript has been corrected accordingly.

Reviewer 2 Report

In this manuscript titled "Numerical Simulation of Mixing Fluid with Ferrofluid in a Magnetic Field by Using the Meshless SPH Method", the authors performed extensive simulation studies on the effect of the number of electrical wires and the Reynolds numbers on the homogeneity of the mixture. 

The experiments are well designed and the conclusions are well supported by the results. I would recommend this manuscript be accepted in the present form with minor language changes. 

Author Response

Response to the Reviewers' Comments

Fluids

Manuscript Title: Numerical Simulation of Mixing Fluid with Ferrofluid in a Magnetic Field by Using the Meshless SPH Method

Authors: M. Abdolahzadeh, A. Tayebi, M. Ahmadinejad, and B. Šarler

Manuscript ID: fluids-1947965

Response to reviewer #2:

All of the reviewers' comments are addressed point by point in the following. Please note that the reviewers' comments are in black, replies to the reviewer are in blue and revisions made to the manuscript are in green.

Remark 2.1:In this manuscript titled "Numerical Simulation of Mixing Fluid with Ferrofluid in a Magnetic Field by Using the Meshless SPH Method", the authors performed extensive simulation studies on the effect of the number of electrical wires and the Reynolds numbers on the homogeneity of the mixture. 

The experiments are well designed and the conclusions are well supported by the results. I would recommend this manuscript be accepted in the present form with minor language changes. 

Reply 1.18: The authors wish to thank the reviewer for his/her dedication to reviewing our manuscript and his positive response.

Reviewer 3 Report

The paper reports the results of a simulated model. The methods and software used (except equations) are not well described in the paper as well as the SW used for comparison. The results are accurately reported but the input data are not defined. Then it is difficult to understand the magnitude of the effect. The problem could be difficult to be reproduced since any input quantities is defined. What is typical length and diameter of a typical channel? The paper has to be extensively revised including any data suitable to understand the phenomena.

Please, define all acronym at their first usage. E.g. SPH

In the abstract at line 5 please clarify that the wires that the authors investigate are  parallel wires (correct?). In Fig. 1 what is UL?

Line 16 please define the mixing object in the sentence ‘mixing process at,,,

Line 25 please, check they are essential n some of applications

Line 84 please clarify the object of investigation continuity, momentum, and mass transport equations, of what quantity? What is Xi in equations (1)-(3)? And Mn?

Line 106 What represent the magnetic number?

Line 108 D is a diameter or the edge of a square channel?

Eq(8) what quantity represent?

Eq (9)-(10) please, describe accurately all quantities used in these equations and the resulting quantity.

Line 129 please, specify in the frame of the works what scalar field A or a vector field A of particle represent.

E(11)-(18) please, define accurately all the quantities

Line 152 Please define or give a reference for Dummy particle method

Line 156 there is a [?] eference. Please, put the number.

Eq(18) Please, what are v, g, alpha..

Line 165 Please, describe the Lattice Boltzmann method and underline the difference with the other used by authors.

Fig. 3 and 4 Please, add the position of the wire and if it is possible the magnetic field distribution. Why the case Re = 15 is representative?

Line 184 Please, define the two fluid thar are mixing.

Line 187 specific desired time Please, give typical range.

Line 188 Please, support the sentence the particle dispersion at the outlet of the channel is improved, which favourable influences the mixing index.with data.

Line 193 Authors state that for the case of two and three wires are better than in the case of one wire. Please, represent the effect of one or more wire in the magnetic field distribution. Please, give the characteristics of the electrical current used in the three wires. Frequency, amplitude, the same intensity in all the wires? The inter-wire distance was analyzed by the authors? What is the influence of this parameter? And the distance by the tube was analyzed? The nanoparticles magnetic characteristic could influence the mixing effect and the shape of the magnetic field? How is the movement of the fluid obtained? By the magnetic field or by a pump and the magnetic field affect only the NP mixing? It is not clear. Why higher Reynolds numbers require more wire? are the discussed results affected by the nanoparticle concentrations? How? What is the typical behavior of a fluids with Re =1 and Re =100?

Fig 6: please labels are invisible.

Authors performed an experimental validation?

Author Response

Response to the Reviewers' Comments

Fluids

Manuscript Title: Numerical Simulation of Mixing Fluid with Ferrofluid in a Magnetic Field by Using the Meshless SPH Method

Authors: M. Abdolahzadeh, A. Tayebi, M. Ahmadinejad, and B. Šarler

Manuscript ID: fluids-1947965

Response to reviewer #3:

The authors wish to thank the reviewer for his/her dedication to review our manuscript and for his/her very helpful comments that contributed to the quality of our publication.

All of the reviewers' comments are addressed point by point in the following. Please note that the reviewers' comments are in black, replies to the reviewer are in blue and revisions made to the manuscript are in green.

Remark 3.1: The paper reports the results of a simulated model. The methods and software used (except equations) are not well described in the paper as well as the SW used for comparison. The results are accurately reported but the input data are not defined. Then it is difficult to understand the magnitude of the effect. The problem could be difficult to be reproduced since any input quantities is defined. What is typical length and diameter of a typical channel? The paper has to be extensively revised including any data suitable to understand the phenomena.

Reply 3.1: The first author apologizes for the problem that occurred when running Latex. In the Latex source file, a command, \begin{linenomath*}, caused the converted pdf file not to display part of the manuscript, part of the introduction (section 1) and physical models description (section 2). the physical description was presented in section 2. Therefore, the latex problem was fixed and section 2 was added as follows:

Revision 3.1:‌ A scheme of the considered micromixer is depicted in Fig. 1. The microchannel depth is assumed to be large enough to ignore variations of the flow field properties perpendicular to the x-y plane. Several experimental studies proved the possibility of studying the mixing process in two dimensions [1,2]. Therefore, considering 2D models of micromixers provides a reasonably good approximation to some of their designs [3]. Dimensions of microchannel here are specified based on a unit length, UL. As Fig. 1 shows, the two fluids enter the microchannel with the same velocity, Uin . The miscible fluids have the same viscosity and density, whereas their concentrations are Co = 0.0 and C1 = 1.0, at the entrance of the microchannel. The three electrical wires generating the magnetic field are located at fixed positions, as depicted in (Fig. 1).

Figure 1. Schematic of physical models of micromixers.

Remark 3.2: Please, define all acronym at their first usage. E.g. SPH

Reply 3.2: The acronyms were defined in the correct place.

Remark 3.3: In the abstract at line 5 please clarify that the wires that the authors investigate are  parallel wires (correct?). In Fig. 1 what is UL?

Reply 3.3: The "electrical parallel wires" was clarified in the abstract. UL is the unit length as was mentioned in section 2.

Revision 3.3:‌ After validating the computational approach, the influence of the number (from one to three) of parallel electrical wires positioned perpendicular to the microchannel on the mixing efficiency is studied for the first time.

Remark 3.4: Line 16 please define the mixing object in the sentence' mixing process at,,,'

Reply 3.4: This sentence was rewritten as follows:

Revision 3:4:‌ The study of the mixing process of fluids in the submillimeter scales represents an interesting research subject due to the design of different microfluidic systems.

Remark 3.5: Line 25 please, check 'they are essential n some of applications'

Reply 3.5: This sentence was rewritten as follows:

Revision 3.5: , they are essential in some of the applications...

Remark 3.6: Line 84 please clarify the object of investigation' continuity, momentum, and mass transport equations,' of what quantity? What is Xi in equations (1)-(3)? And Mn?

Reply 3.6: The purpose of studying the governing equations is to evaluate the performance of micromixers using the mixing index defined in line 95 . xi is the direction of the Cartesian coordinates, which in the present case is a Cartesian 2D system (X,Y), Fig 1. Mn is the dimensionless magnetic number which is defined in line 112 . However, the sentences were reworded as follows:

Revision 3.6a: In this section, the conservation laws, i.e. the continuity, momentum, and mass transport equations, as the governing equations to study the effects of different magnetic fields on the mixing process of ferrofluid, using a mixing index defined subsequently, are presented as:

Revision 3.6b:‌ where t, xi, ρ, ui , p, ν, Mn, H, C, and αc are the time, direction of the 2D Cartesian system (here are X and Y), density of the fluid, velocity components, pressure, kinematic viscosity, dimensionless magnetic number, the magnitude of the magnetic field, concentration, and mass diffusivity, respectively.

Remark 3.7: Line 106 What represent the 'magnetic number'?

Reply 3.7: The magnetic number is the most important parameter in the magnetism problems, which determines the influence of ferro-hydrodynamics. In other words, the magnetic number expresses the ratio of the magnetic forces and the inertial forces [4, 5]

Remark 3.8: Line 108 D is a diameter or the edge of a square channel?

Reply 3.8: D is the width of the channel shown in Fig 1.

Remark 3.9: Eq(8) what quantity represent?

Reply 3.9: SPH is a weakly compressible method. Therefore, the mixed fluids in the present study are considered weekly compressible fluids. The system of equations, composed of relations 3.1, 3.2, and 3.3, is closed by a relationship between density and pressure. Assuming the weakly compressible condition causes the pressure field to come out from a suitable equation of state. Various equations of state have been used to calculate the pressure field from the local density and temperature. In the present study, Morris' equation of state is used.

Remark 3.10: Eq (9)-(10) please, describe accurately all quantities used in these equations and the resulting quantity.

Reply 3.10: Revised as suggested.

Revision 3.10a:‌ To quantify the mixing performance of micromixers, a mixing index, based on the standard deviation of concentration, can be defined as…

Revision 3.10b:‌ where N, Cj , and Cmean are, respectively, total number of particles, concentration of particle j, and the mean concentration (here Cmean = 0.5). The mixing index can be evaluated from 0, unmixed state, to 1, full mixed state. Eq. 15 ...

Revision 3.10c:‌ higher values of mixing efficiency ηmix indicate greater homogeneity of the mixed fluids and thus higher performance of the micromixer.

Remark 3.11: Line 129 please, specify in the frame of the works what scalar field A or a vector field A of particle represent.

Reply 3.11: The pressure and concentration are represented as scalar fields, and the velocity and magnetic field as vector fields. Therefore the sentence was revised as follows:

Revision 3.11:‌ The function of interpolation for a scalar field A , the pressure and concentration, or a vector field A, the velocity and magnetic field, of particle a ...

Remark 3.12: E(11)-(18) please, define accurately all the quantities

Reply 3.12: The authors noted that w'ab, the derivative of the kernel function, and g, gravitational acceleration, were not defined. However, some parameters such as αc were defined in the first stage of the mention.

Revision 3.12: in which eab, w'ab and Ba are respectively the unit vector between particles a and b, the derivative of the kernel function, and ...

Remark 3.13: Line 152 Please define or give a reference for 'Dummy particle method'

Reply 3.13: The related reference was added.

Revision 3.13: In this work, the Dummy particle method [6] is used for wall boundaries, and the open channel flow conditions are considered for inflow and outflow boundaries [7].

Remark 3.14: Line 156 there is a [?] eference. Please, put the number.

Reply 3.14: The related reference was corrected.

Remark 3.15: Eq(18) Please, what are v, g, alpha.

Reply 3.15: the definition of ν, the kinematic viscosity, was mentioned in line 101, g is gravity acceleration.

Revision 3.15:… between two particles, g is gravity acceleration and ...

Remark 3.16: Line 165 Please, describe the 'Lattice Boltzmann method' and underline the difference with the other used by authors.

Reply 3.16: The following paragraph has been added to describe LBM and its difference from SPH.

Revision 3.16:‌ LBM, which originated in lattice gas automata, is another class of computational fluid mechanics that uses streaming and collision (relaxation) processes to simulate a fluid density on a lattice rather than directly implementing the Navier-Stokes equations. In contrast, SPH is a meshless method that directly solves the Navier-Stokes equations to simulate fluid flow in a Lagrangian framework.

Remark 3.17: Fig. 3 and 4 Please, add the position of the wire and if it is possible the magnetic field distribution. Why the case Re = 15 is representative?

Reply 3.17: The effects of magnetic fields are considered in the Navier-Stockes equation. We have not considered the schemes of magnetic fields in the contours. The location of the magnetic fields is shown in Fig. 1.

The contours of the Reynolds numbers are not very different, which requires additional description. Therefore, the authors decided to present the contours at Re = 15 in the manuscript as an example of the results for the other Reynolds numbers.

Remark 3.18: Line 184 Please, define the two fluid thar are mixing.

Reply 3.18: In the present work, the authors used dimensionless numbers instead of using special fluids. However, the following sentence has been added:

Revision 3.18: ...between two fluids and ferrrofluid...

Remark 3.19: Line 187 'specific desired time' Please, give typical range.

Reply 3.19: The mixing index at the outlet reaches a steady state. The results have been reported at t*= 0.014, where t* = tUin/UL is dimensionless time. Therefore, the sentence has been rewritten as:

Revision 3.19: It is worth mentioning that the results are presented at t*= 0.014, where t* = tUin/UL is the dimensionless time.

Remark 3.20: Line 188 Please, support the sentence 'the particle dispersion at the outlet of the channel is improved, which favourable influences the mixing index.' with data.

Reply 3.20: the mixing efficiency for one, two, and three wires at Re =14 are, respectively, 61.85, 72.22, and 77.16. Therefore, the following sentence has been added:

Revision 3.20: The mixing efficiencies at Re=15 for one, two and three wires are 61.85 %, 72.22 % and 77.16 % , respectively, confirming this fact.

Remark 3.21: Line 193 Authors state that 'for the case of two and three wires are better than in the case of one wire'. Please, represent the effect of one or more wire in the magnetic field distribution.

Reply 3.21: The authors mentioned that "for the interval of 0 < Re ≤ 20, the mixing efficiencies for the case of two and three wires are better than in the the case of one wire" not for all the Reynolds number. The effects of different numbers of electrical wires on the mixing process are shown in Fig. 5. As this figure shows, two and three wires provide higher mixing efficiency compared to one wire.

Remark 3.22: Please, give the characteristics of the electrical current used in the three wires. Frequency, amplitude, the same intensity in all the wires?

Reply 3.22: It is worth mentioning that the present simulation was based on dimensionless numbers, including the Reynolds number, the Schmidt number, and the magnetic number. The magnitudes of these parameters were given in line 189. The magnetic parameters are the magnetic number Mn = 1000, the saturation magnetization Ms = 60 A/m and the magnetization at the location of the wires Mo = 60.

Remark 3.23: The inter-wire distance was analyzed by the authors? What is the influence of this parameter? And the distance by the tube was analyzed? The nanoparticles magnetic characteristic could influence the mixing effect and the shape of the magnetic field?

Reply 3.23: In the present work, the location of the wires was chosen to be the same for all case studies. The number of electric wires and Reynolds number were considered as factors for the study.

Remark 3.24: How is the movement of the fluid obtained? By the magnetic field or by a pump and the magnetic field affect only the NP mixing? It is not clear.

Reply 3.24: In section 2 of the manuscript, the authors mention that both the fluid and the ferrofluid enter the microchannel at the same velocity. Therefore, the motion of both the fluid and the ferrofluid is affected by both the inlet velocity and the magnetic fields.

Remark 3.25: Why higher Reynolds numbers require more wire?

Reply 3.25: At higher Reynolds numbers, increasing the velocity of the mixing fluids shortens the contact time of the fluids, which is a factor of mixing efficiency. Increasing the magnetic fields intensifies the generation of vortex shedding, which increases the contact time.

Remark 3.26: are the discussed results affected by the nanoparticle concentrations? How?

Reply 3.26: Particle diameter was not considered in the study. The ferrofluid concentration in the initial state is c=1, while the concentration changes during the mixing process based on the concentration equation. In the present study, ferrofluid was considered as a colloidal liquid consisting of nanoscale particles suspended in a carrier fluid.

Remark 3.27: What is the typical behavior of a fluids with Re =1 and Re =100?

Reply 3.27: At Re = 1, increasing the number of electric wires increases vortex shedding, resulting in an improvement in homogeneity. However, at Re 100, the contact time decreases due to the higher velocity value during the mixing process. However, the vortex shedding caused by the magnet wires increases the interface between the two fluids, resulting in better mixing efficiency.

Remark 3.28: Fig 6: please labels are invisible.

Reply 3.28: Thank you for the comment. The labels have been corrected.

Remark 3.29:Authors performed an experimental validation?

Reply 3.29: The SPH code has not been experimentally validated for the same case. However, the authors validated their SPH code experimentally with a dam-breaking problem for Newtonian and non-Newtonian fluids [8]. Other numerical validations for the current code were also presented in [7, 9].

References

[1] Gobby, D.; Angeli, P.; Gavriilidis, A. Mixing characteristics of T-type microfluidic mixers. Journal of Micromechanics and microengineering 2001, 11, 126.

[2] Erickson, D.; Li, D. Microchannel flow with patchwise and periodic surface heterogeneity. Langmuir 2002, 18, 8949–8959.

[3] Bedram, A.; Moosavi, A.; Hannani, S.K. Analytical relations for long-droplet breakup in asymmetric T junctions. Physical Review E 2015, 91, 053012.

[4] Senel, P. E. L. İ. N., and M. Tezer-Sezgin. "DRBEM solution of biomagnetic fluid flow and heat transfer in cavities-CMMSE2016." Journal of Mathematical Chemistry 55.7 (2017): 1407-1426.

[5] Larimi, M. M., et al. "Forced convection heat transfer in a channel under the influence of various non-uniform transverse magnetic field arrangements." International Journal of Mechanical Sciences 118 (2016): 101-112.

[6] Lee, E.S.; Moulinec, C.; Xu, R.; Violeau, D.; Laurence, D.; Stansby, P. Comparisons of weakly compressible and truly incompressible algorithms for the SPH mesh free particle method. Journal of Computational Physics 2008, 227, 8417 8436.

[7] Abdolahzadeh, M.; Tayebi, A.; Mansouri Mehryan, M. Numerical simulation of mixing in active micromixers using SPH. Transport in Porous Media 2022, pp. 1–18.

[8] Abdolahzadeh, Mohsen, Ali Tayebi, and Pourya Omidvar. "Mixing process of two-phase non-newtonian fluids in 2D using smoothed particle hydrodynamics." Computers & Mathematics with Applications 78.1 (2019): 110-122.

[9] Abdolahzadeh, Mohsen, Ali Tayebi, and Pourya Omidvar. "Thermal effects on two-phase flow in 2D mixers using SPH." International Communications in Heat and Mass Transfer 120 (2021): 105055.

Round 2

Reviewer 3 Report

The paper was improved. Nevertheless give typical size of UL and also typical curent: direct current or time varying with what amplitude? What current amplitu is need to obtain the magnetic flux density at line 189?

Please, add all information suitable to implement the model

Fig. 4 what is the particle contour?

Please, add relevant comments in the result part.

Author Response

Response to the Reviewers' Comments

Fluids

Manuscript Title: Numerical Simulation of Mixing Fluid with Ferrofluid in a Magnetic Field by Using the Meshless SPH Method

Authors: M. Abdolahzadeh, A. Tayebi, M. Ahmadinejad, and B. Šarler

Manuscript ID: fluids-1947965

Response to reviewer #3:

The authors wish to thank the reviewer for his/her dedication and for once again carefully reviewing our manuscript. The comments (in black) are addressed point by point in the following. Our replies to the reviewer are in blue and revisions made to the manuscript are in green.

Remark 3.1: The paper was improved. Nevertheless give typical size of UL and also typical current: direct current or time varying with what amplitude? What current amplitude is need to obtain the magnetic flux density at line 189?

Reply 3.1: The UL is a unit of length that can be set with any number under the limitation of dimension numbers, Re, Sc, Mn used in this study, here UL=1. It is worth mentioning that the authors reported the result with dimensionless quantities. Stating the results with dimensionless numbers is helpful for the user to perform the same case study with other quantities under the limitation of dimensionless numbers. Therefore, the user can set up the case study according to the following procedure:

The magnetic field induction, B0, can be calculated using Mn; here, Mn=1000. Then the magnetic strength, H0, can be obtained by defining B0. It should be noted that H0 is the magnetic strength at the minimum distance between the bottom wall of the microchannel and the location of the wires, as mentioned in the manuscript in line 115. Then the current I can be calculated with the value of H0 at the defined coordinate.

Remark 3.2: Please, add all information suitable to implement the model

Reply 3.2: The required additional information to set up the case study has been provided as a response in the previous comment.

Remark 3.3: Fig. 4 what is the particle contour?

Reply 3.3: It is worth noting that SPH is a full Lagrangian particle method which makes it possible to track each fluid particle motion during the time. One of the advantages of Lagrangian analysis is that the mixing process caused by fluid motion and molecular diffusion are studied simultaneously. Therefore, this contour provides the particle distribution and mixing of fluid particles caused by fluid motion.

Remark 3.4: Please, add relevant comments in the result part.

Reply 3.3: The authors believe we have addressed all of the reviewer's comments, and the appropriate revisions have been incorporated into the manuscript.
